# Infection Heterogeneity and Microbiota Differences in Chicks Infected by *Salmonella enteritidis*

**DOI:** 10.3390/microorganisms9081705

**Published:** 2021-08-11

**Authors:** Shu Wu, Guanglei Cong, Qianyun Zhang, Hong Yao, Zhenxin Wang, Kelang Kang, Xi He, Shourong Shi

**Affiliations:** 1College of Animal Science and Technology, Hunan Agricultural University, Changsha 410128, China; wushu223759@163.com (S.W.); kangkelang@126.com (K.K.); 2Department of Feed and Nutrition, Poultry Institute, Chinese Academy of Agricultural Sciences, Yangzhou 225125, China; MZ120181027@yzu.edu.cn (G.C.); zhangqy1120@163.com (Q.Z.); Ravenpeach@163.com (H.Y.); wangzhenxin1995@163.com (Z.W.); 3Institute of Effective Evaluation of Feed and Feed Additive (Poultry Institute), Ministry of Agriculture, Yangzhou 225125, China; 4Jiangsu Co-Innovation Center for the Prevention and Control of Important Animal Infectious Disease and Zoonose, Yangzhou University, Yangzhou 225009, China

**Keywords:** *Salmonella enteritidis*, heterogeneity, cecal microbiome, intestinal barrier, *Desulfovibrio_piger*

## Abstract

This study was conducted to compare the infection heterogeneity and cecal microbiota in chicks infected by *S. enteritidis*. Forty-eight 8-d-old female Arbor Acres chicks were challenged with *S. enteritidis* and euthanized 24 h later. The eight chicks with the highest *Salmonella* tissue loads were assigned to group S (*S. enteritidis*-susceptible), and the eight chicks with the lowest *Salmonella* tissue loads were assigned to group R (*S. enteritidis*-resistant). Chicks in group S showed a higher liver index (*p* < 0.05), obvious liver lesions, and an decreasing trend for the villus height-to-crypt depth ratio (*p* < 0.10), compared with those in group R. Gene expression of *occludin*, *MUC2*, and *IL10* was higher, whereas that of *iNOS* and *IL6* was lower (*p* < 0.05), in chicks of group R relative to those in group S. Separation of the cecal microbial community structure has been found between the two groups. The *S. enteritidis*-susceptible chicks showed higher abundance of pathogenic bacteria (*Fusobacterium* and *Helicobacter*) in their cecal, while *Desulfovibrio_piger* was enriched in the cecal of *S. enteritidis*-resistant chicks. In summary, chicks showed heterogeneous responses to *S. enteritidis* infection. Enhanced intestinal barrier function and cecal microbiota structure, especially a higher abundance of *Desulfovibrio_piger*, may help chicks resist *S. enteritidis* invasion.

## 1. Introduction

*Salmonella* is a major foodborne pathogen of global importance, which has led to large numbers of deaths in humans and caused economic losses in animal husbandry [1]. Among the more than 2500 identified *Salmonella enterica* serotypes, *Salmonella enteritidis* (*S. enteritidis*) is the most frequently spread from animals to humans globally [2]. *S. enteritidis* has caused occasional epidemic outbreaks around the world, such as in China [3], South Africa [4] and the United States [5]. Poultry are the primary *S. enteritidis* host, and the percent prevalence of *S. enteritidis* in chicken meat is strongly positively correlated (r = 0.804, *p* ≤ 0.01) with the incidence of human illnesses caused by this serotype [6]. These observations highlight the importance of studying *S. enteritidis* infection in poultry for reasons associated with both public health and poultry production.

Host susceptibility to pathogen infection is frequently heterogeneous [7], as demonstrated by the phenomenon of the median lethal dose (lethal dose 50 [LD_50_]), which describes the microbe dose that will kill only 50% of a test population [8]. Poultry infected with *S. enteritidis* may suffer systemic infection that can potentially lead to death, or may evolve into a long-term asymptomatic carrier-state [9]. Several studies have confirmed that heterogeneous responses to *Salmonella* infection can be partly explained by the genetic background and immune function status of the host [10,11]. However, numerous studies have also reported the phenomenon of heterogeneous bacterial shedding (super-shedders and low-shedders) in genetically homogeneous host populations [12,13], suggestive of the existence of additional factors that can influence the susceptibility and resistance of individuals to *Salmonella* colonization. Over recent years, the composition of the intestinal microbiota has been increasingly associated with heterogeneous host responses to pathogen infection [14,15,16].

The intestinal microbiota comprises a complex bacterial community and maintaining a mutually beneficial balance between the host and the gut microflora is very important for human health [17,18]. Intestinal dysbiosis can promote or even directly lead to a variety of conditions, including inflammatory diseases, colon cancer, and autoimmune disorders [19]. Pathogen infection is also closely related to the intestinal microbiota. Pathogen infection can lead to an imbalance in the intestinal environment, where pathogen growth is favored over that of probiotics [20,21,22]. Conversely, the gut microbiota can help inhibit pathogen colonization [23,24]. Although various mechanisms through which gut microbiota can protect the host against intestinal infection have been described, it remains unclear whether the heterogeneous responses of poultry to *S. enteritidis* infection are related to subtle changes in gut microbiota composition. In this study, we investigated the infection of *S. enteritidis*-susceptible and -resistant chicks from the aspects of tissue lesions, intestinal health and inflammatory response, and analyzed their cecal microbiota differences.

## 2. Materials and Methods

### 2.1. Ethical Statement

The study was conducted according to the Regulations of the Experimental Animal Administration issued by the State Committee of Science and Technology of the People’s Republic of China. The animal use protocol was approved by the Animal Care and Use Committee of the Poultry Institute, Chinese Academy of Agriculture Science (No. CNP20201030).

### 2.2. Animal Management

Forty-eight 1-d-old Arbor Acres (AA) broiler chicks were obtained from Jiangsu Jinghai Poultry Industry Group Co., Ltd. (Nantong, Jiangsu, China). Cloacal swab tests [25] were carried out immediately after hatching to exclude *Salmonella* infection. The chicks were reared in cages with a wire screen floor. Water and feed were provided ad libitum, with the photoperiod set at 24 L throughout the study. The temperature in the broiler house during the first week ranged from 32 to 35 °C, and was then decreased by 1 °C d^−1^ until reaching the final temperature of 30 °C on d 9. The diet of the chicks, without antibiotics or anticoccidial drugs and negative for *Salmonella*, was formulated to meet or slightly exceed all nutrient requirements (NRC, 1994) and was prepared at the Poultry Institute, Chinese Academy of Agriculture Science. The nutrient composition is shown in Table 1.

### 2.3. Challenge with S. enteritidis

Forty-eight chicks were orally gavaged with 1 × 10^9^ colony forming unit (CFU) of *S. enteritidis* at d 8 as previously described [1,26,27]. Briefly, the *S. enteritidis* strain CMCC(B)50041 (Bei Na Chuanglian Biotechnology Co., Ltd., Suzhou, Jiangsu, China) were grown in modified Martin medium (Qingdao-Hope Biotechnology Co., Ltd., Qingdao, China) overnight at 37 °C with constant shaking. Before inoculation, the bacteria were washed with PBS, and serially diluted to a concentration of 1 × 10^9^ CFU/mL based on the optical density at 600 nm measured by a microplate reader (Infinite M200 Pro, Tecan, Switzerland). The bacterial stock was kept on ice before infection. After infection, the same bacterial stock was plated on xylose lysine desoxycholate (XLD) agar (Qingdao-Hope Biotechnology Co., Ltd., Qingdao, China) to verify the CFU accuracy.

### 2.4. Sample Collection

At 24 h post infection (hpi) (age = 9 d), all the chicks were euthanized by severing the jugular vein. The body weight, liver weight and spleen weight were measured, and index of liver and spleen were calculated. About 0.2~0.4 g of liver and a half of the spleen were collected aseptically from each chick and stored at 4 °C for *Salmonella* load quantification. The cecal content was collected, frozen in liquid nitrogen, and stored at −80 °C for 16S rRNA sequencing analysis. Small segments of the rest liver, spleen, and jejunum were collected and immediately fixed in a 10% formaldehyde solution for histopathological examination. In addition, cecal tonsil and segments of jejunum were collected, frozen in liquid nitrogen, and stored at −80 °C for quantification of target gene mRNA levels.

### 2.5. Salmonella Load Measurement and Sample Grouping

To determine the *Salmonella* loads in the liver and spleen, the samples were weighed and diluted in 3 mL of sterile PBS. Then, the samples were homogenized for 120 s at 60 Hz using a SCIENTZ-48 homogenizer (Ningbo Xingzhi Biotechnology Co., Ltd., Ningbo, China). A total of 50 μL of the homogenate liquid from the samples was plated on XLD agar and incubated for 24 h at 37 °C. According to the result of spleen Salmonella loads (log_10_CFU/g), we select log_10_CFU/g > 4.400 as the cut-off value for S. Enteritidis-resistant chicks, and log_10_CFU/g < 2.700 as the cut-off value for *S. enteritidis*-susceptible chicks. According to the cut-off values, eight chicks were assigned to group S, and eight chicks were assigned to group R (Appendix A).

### 2.6. Liver and Spleen Histopathology and Intestinal Morphology Determination

Tissue histopathology and intestinal morphology of chicks from both groups were determined as previously described [28,29]. Briefly, small segments of liver, spleen, and middle jejunum were fixed in 10% buffered formaldehyde (pH 7.2) and dehydrated via an ascending ethanol gradient. After xylene clearing, the samples were embedded in paraffin and processed into 5-µm-thick slices followed by mounting and hematoxylin-eosin (HE) staining. Inflammatory infiltration and general damage in the liver and spleen, as well as the villus height (VH) and crypt depth (CD) of the jejunum, were observed and measured under a fluorescence microscope (DM4000B, Leica Microsystems, Wetzlar, Germany). The ratio of the villus height-to-crypt depth (VCR) was also calculated. Histopathological images of liver or spleen were scored by a pathology professional who did not know the experimental group according to the number of inflammatory cell nodules and the degree of cell degeneration and necrosis. The score from normal to severe lesions was 0~4.

### 2.7. RNA Isolation and Quantitative Real-Time PCR

Total RNA was extracted from the cecal tonsil or middle jejunum of birds from both groups using an RNAsimple Total RNA Kit (Tiangen Biotech Co., Ltd., Beijing, China) following the manufacturer’s instructions. RNA concentration and purity were determined by measuring the absorbance at 260 and 280 nm using a NanoDrop 2000 spectrophotometer (Thermo Fisher Scientific, Rockford, IL, USA), and RNA quality was assessed by agarose gel electrophoresis. Total RNA was reverse-transcribed using the FastKing gDNA Dispelling RT SuperMix Kit (Tiangen Biotech Co., Ltd.) in accordance with the manufacturer’s instructions. Reverse transcription was performed at 42 °C for 15 min followed by heat inactivation for 3 min at 95 °C. The cDNA was stored at −20 °C until further use. Real-time quantitative PCR was performed in a StepOnePlusTM Real-Time PCR System (Applied Biosystems, Foster City, CA, USA) following optimized PCR protocols using a SuperReal PreMix Plus (SYBR Green) Kit (Tiangen Biotech Co., Ltd.). The protocol consisted of an initial denaturation step at 95 °C for 15 min, followed by 40 cycles of 10 s denaturation at 95 °C and 30 s annealing/extension at 60 °C, with a final step at 95 °C for 15 s. The primers for inducible nitric oxide synthase (*iNOS*), interferon-gamma (*IFNG*), tumor necrosis factor-alpha (*TNFA*), interleukin 1 beta (*IL1B*), *IL6*, *IL8*, *IL10*, *occludin*, *claudin*, zonula occluden 1 (*ZO-1*), mucin 2 (*MUC2*), and glyceraldehyde-3-phosphate dehydrogenase (*GAPDH*) are listed in Table 2. The ΔΔCt method was used to estimate mRNA abundance. *GAPDH* was used as the internal reference gene, and the mRNA expression of target genes was normalized to that of *GAPDH*.

### 2.8. DNA Extraction and Sequencing Library Construction

Genomic DNA was extracted from homogenized cecal content using CTAB method [30] and stored at −20 °C. DNA concentration and purity were assessed by 2% agarose gel electrophoresis and diluted to 1 ng/μL using sterile water. The V4 region of the bacterial 16S rRNA gene was PCR amplified using the barcoded 515F/806R primer pair [31]. Amplicons consisting of around 400–450 bp were extracted and used for further analysis [32,33]. PCR products were purified using the QIAquick Gel Extraction Kit (Qiagen Inc., Santa Clara, CA, USA). Sequencing libraries were generated using the Illumina TruSeq^®^ DNA PCR-Free Sample Preparation Kit (Illumina, San Diego, CA, USA) following the manufacturer’s recommendations. After Qubit-based quantification and library qualification, the library was subjected to sequencing at Novogene Co., Ltd. (Beijing, China) using the Illumina NovaSeq6000 platform.

### 2.9. Quality Filtering and Sequence Analysis

Raw Illumina paired-end reads were trimmed of barcodes and primers and combined using Flash software (V1.2.7, http://ccb.jhu.edu/software/FLASH/, accessed on 1 February 2021) with default parameters [34]. The obtained raw sequence data were quality-filtered using QIIME V1.9.1 (http://qiime.org/scripts/split_libraries_fastq.html, accessed on 1 February 2021) to obtain effective tags [35]. OTU were assigned at 97% identity using Uparse V7.0.1001 (http://www.drive5.com/uparse/, accessed on 1 February 2021) based on the effective tags [36]. OTU taxonomic information was annotated by RDP Classifier using a 0.8~1 confidence threshold for taxonomic assignment [37,38]. Alpha and beta diversity and the significance of taxonomic differences between samples were estimated by QIIME (V1.9.1) and linear discriminant analysis effect size (LEfSe) as previously described [38,39,40].

### 2.10. Statistical Analysis

Statistical analyses were carried out with SPSS for Windows V22.0 (SPSS Inc., Chicago, IL, USA). Differences between two groups were tested by independent samples *t*-tests and the Wilcoxon rank-sum test. Data are expressed as means ± SEM. A *p*-value < 0.05 was considered statistically significant [26].

## 3. Results

### 3.1. Body Weight, Tissue Index, and Salmonella Loads of S. enteritidis-Susceptible and -Resistant Chicks

The differences in body weight, tissue indices, and *Salmonella* loads between *S. enteritidis*-susceptible and -resistant chicks are summarized in Table 3. The liver index, liver *Salmonella* load, and spleen *Salmonella* load of *S. enteritidis*-susceptible chicks (group S) were higher than those of *S. enteritidis*-resistant chicks (group R) (*p* < 0.05) at 24 hpi; however, there was no significant difference in body weight or spleen index between the two groups.

### 3.2. Liver and Spleen Histopathology of S. enteritidis-Susceptible and -Resistant Chicks

The differences in liver and spleen histopathology between *S. enteritidis*-susceptible and -resistant chicks are shown in Figure 1. There were only slight pathological changes in the livers of the birds in group R, with only limited infiltration of heterophilic cells and lymphocytes being observed around some of the blood vessels (Figure 1a,e). In contrast, chicks in group S showed obvious lesions in their livers, including numerous lymphocyte nodules and infiltrated heterophilic cells, as well as pyknosis of liver nuclei (Figure 1b,e). No obvious pathological changes were found in the spleens of chicks in the two groups (Figure 1c,d).

### 3.3. Intestinal Morphology and Barrier Function of S. enteritidis-Susceptible and -Resistant Chicks

The differences in jejunum morphology between the *S. enteritidis*-susceptible and -resistant chicks are summarized in Table 4. Although the chicks in group R showed lower CD, higher VH, VCR, and muscle thickness (MT), the differences between the two groups were not statistically significant. Only the VCR showed a higher trend in group R (*p* < 0.10).

We further investigated the differences in barrier function between *S. enteritidis*-susceptible and -resistant chicks by comparing their expression of the *claudin*, *occludin*, *ZO-1*, and *MUC2* genes in the jejunum. As shown in Figure 2a, the expression of occludin and *MUC2* was lower in the jejunum of *S. enteritidis*-susceptible chicks than in that of *S. enteritidis*-resistant chicks (*p* < 0.05). No statistically significant differences in claudin or *ZO-1* expression were found between the two groups.

### 3.4. Expression of Inflammatory Cytokine-Related Genes in S. enteritidis-Susceptible and -Resistant Chicks

The gene expression of *iNOS*, *IFNG*, *TNFA*, *IL1B*, *IL6*, *IL8*, and *IL10* in the cecal tonsil of both groups of chicks are shown in Figure 2b. Compared with group R, the expression of the genes encoding the proinflammatory factors *iNOS* and *IL6* were markedly higher in the chicks of group S, whereas that of *IL10*, encoding an anti-inflammatory factor, was significantly lower (*p* < 0.05).

### 3.5. Composition and Diversity of Cecal Microorganisms in S. enteritidis-Susceptible and -Resistant Chicks

A total of 857,129 effective reads were obtained from 16 cecal digesta samples (8 samples per group), and these reads were assigned to 2458 operational taxonomic units (OTU) (Appendix A). Each sample contained 53,571 ± 1632 (mean ± SEM) effective reads and 694 ± 46 (mean ± SEM) OTU on average. Good’s coverage indices were greater than 99.5% for all the cecal digesta samples (Appendix A) and rarefaction curves based on the observed OTU reached a plateau (Appendix A), both indicating that sequencing coverage was sufficient to represent all OTU present in the samples.

No significant difference was found in microbial community richness and diversity between the *S. enteritidis*-susceptible and -resistant chicks by alpha diversity analysis, including ACE, Chao1, PD_whole_tree, Shannon, and Simpson indices (Appendix A). However, beta diversity analysis indicated there was a separation of the cecal microbial community structure between the *S. enteritidis*-susceptible and -resistant chicks, as illustrated by principal component analysis (PCA), non-metric multidimensional scaling (NMDS), and unweighted pair-group method with arithmetic mean (Figure 3). The beta diversity of group R was lower than that of group S as calculated by binary_jaccard and unweighted_unifrac (Appendix A).

Data for the top 10 microbial populations of the cecal bacterial community were analyzed at the phylum level. As shown in Figure 4a, *Bacteroidota*, *Firmicutes*, *Proteobacteria*, and *Actinobacteriota* (59.73% vs. 62.60%, 26.07 vs. 23.97%%, 2.81% vs. 3.08%, and 3.10% vs. 3.36%, for group S vs. group R, respectively) constituted the four dominant phyla in both groups of chicks. Among the top 10 microbial populations, the relative abundance of *Acidobacteriata*, *Campilobacterota*, and *Fusobacteriota* in group R was lower than that in group S (Wilcoxon test, *p* < 0.05) (Figure 4b).

At the genus level, the top 10 genera of the cecal bacterial community (Figure 4c) did not differ significantly between the two groups. Differentiation analysis was also conducted on other identified low-abundance genera. As shown in Table 5, a total of 18 genera showed significantly different abundance between group S and group R. Among them, *Fusobacterium*, *Helicobacter*, *Butyricicoccus*, *Bryobacter*, *Acidothermus*, *unidentified_Chloroplast*, *NK4A214_group*, *Marvinbryantia*, *Burkholderia-Caballeronia-Paraburkholderia*, *Granulicella*, *Puia*, *unidentified_IMCC26256*, *Actinospica*, *Dyella*, and *Nocardia* had higher abundance in group S than those in group R; while *Oribacterium*, *Herbinix*, and *Papillibacter* had lower abundance in group S than that of group R (Wilcoxon test, *p* < 0.05).

To identify differentially abundant biomarkers in *S. enteritidis*-susceptible and -resistant chicks, we employed LEfSe (Figure 5). A cladogram representative of the structure of the microbial communities and their predominant bacteria is shown in Figure 5a. Only taxa with linear discriminant analysis (LDA) values greater than 3 are shown for clarity (Figure 5b). At the phylum level, *Acidobacteriota*, *Campilobacteriota*, *Fusobacteriota*, and *Kapabacteria* were enriched in group S (green circles). At the class level, *Acidobacteria*, *Campilobacteria*, and *Fusobacteriia* were enriched in group S. At the order level, *Campylobacterales*, *Fusobacteriales* and *Kapabacteriales* were enriched in group S and *Veillonellales-Selenomonadales* were prevalent in group R (red circle). At the family level, *Barnesiellaceae*, *Helicobacteraceae*, *Butyricicoccaceae*, and *Fusobacteriaceae* were enriched in group S. Three genera (*Fusobacterium*, *Helicobacter*, and *Butycicoccus*) had higher LDA scores in group S. Two species (*Helicobacter_pullorum* and *Bacteroides_caecicola*) had higher LDA scores in group S; one specie (*Desulfovibrio_piger*) had a higher LDA score in group R.

## 4. Discussion

*Salmonella* can be transmitted horizontally to chickens from contaminated environmental vectors and vertically from infected hens to offspring. In this study, 1-d-old female AA female chicks, with *Salmonella* infection excluded by cloacal swab testing, were reared and challenged under the same conditions, therefore eliminating the influence of genetic background and environment on the experimental results to the greatest extent, and ensured that all the phenotypic results obtained in this study were due to individual differences. *S. enteritidis* mainly colonizes the liver, spleen, and intestine of poultry after infection [41,42], leading to intestinal damage, a decline in growth performance, and even death. Growth performance, pathological changes in organs, *Salmonella* loads, and intestinal morphology are important indicators of the severity of *S. enteritidis* infection. In this study, compared with *S. enteritidis*-resistant chicks, the livers of *S. enteritidis*-susceptible chicks became swollen (Table 3) and displayed salient lesions (Figure 1b). In addition, *Salmonella* loads in the liver and spleen of *S. enteritidis*-susceptible chicks were significantly higher than those of *S. enteritidis*-resistant chicks (Table 3). The VCR showed an increasing trend in chicks of group R than in chicks of group S (Table 4). These results indicated that our grouping scheme, i.e., selecting chicks with differential *S. enteritidis* susceptibility, was appropriate, and confirmed the heterogeneous nature of the response of the birds to *S. enteritidis* infection.

The intestinal mucosal barrier serves as the first line of defense between the host and the luminal environment. Composed of epithelial cells and tight junctions, this barrier can prevent the entry of harmful substances, such as pathogens and toxins, into host tissues, organs, and circulating blood [43]. The intestinal epithelium is involved in the formation of the intestinal mucosal barrier by continuously secreting MUC2 to renew the intestinal mucosal layer. Impaired intestinal mucosal barrier function is a key determinant of the pathogenicity of some intestinal bacteria. Studies have shown that *Salmonella* infection can disrupt the intestinal barrier of broilers, and promoting the expression of tight junction proteins through L-arginine supplementation can alleviate *Salmonella* infection, indicating that there is a negative correlation between intestinal barrier function and the severity of *Salmonella* infection [40]. In this study, we compared the expression of genes encoding tight junction proteins and *MUC2* in *S. enteritidis*-susceptible and *S. enteritidis*-resistant chicks. The results showed that the mRNA expression of *occludin* and *MUC2* in the jejunum of *S. enteritidis*-resistant chicks was significantly higher than that of *S. enteritidis*-susceptible chicks, further supporting that a negative correlation exists between intestinal mucosal barrier function and *S. enteritidis* susceptibility.

Because proinflammatory cytokines are essential for initiating immune responses and eliminating pathogens in the host, we hypothesized that chicks in group R would exhibit higher levels of inflammation than those of group S, therefore explaining the greater resistance of the birds in group R to *S. enteritidis* infection at the same dose of *S. enteritidis* challenge. However, our results showed that there was no significant difference in the expression of most proinflammatory factor-related genes between the two groups. Furthermore, the gene expression of *iNOS* and *IL6* showed the opposite trend to what would be expected, i.e., the expression of both genes was significantly higher in group S than in group R, whereas that of *IL10*, coding for an anti-inflammatory factor, was significantly lower. These results suggested that inflammatory cytokines may play a role in the heterogeneous responses in an unexpected way. Or the higher expression levels of proinflammatory cytokine-related genes may also be considered to be a phenotype of *S. enteritidis*-susceptible chicks, which is consistent with the results of the histopathological analysis of liver tissue. In addition, although *iNOS* is believed to help cells resist bacterial invasion through the production of a large amount of NO, which serves as an antibacterial [44], it is notable that the relationship between NO and *Salmonella* in the host may not be merely antagonistic. It has been reported that *Salmonella* needs NO as a nitrogen source for nitrate respiration, and a low NO concentration is indispensable for promoting *Salmonella* growth [45]. This may also explain why the invasion of *S. enteritidis* in birds of group S was more severe, but their expression of the *iNOS* gene were higher in our research.

In the chicken, the intestinal microbiota is composed of complex microbial communities that are involved in digestion and metabolism, the regulation of intestinal cells, vitamin synthesis, and the development and regulation of the host immune system [46]. There is also accumulating evidence indicating that the intestinal microbiota profoundly influences the pathogenicity of *S. enteritidis* [24]. Because the cecum is the most densely colonized microbial habitat in the chicken [47], we systematically compared the cecal microbial composition of chicks from the different *S. enteritidis* susceptibility groups. Alpha diversity refers to the richness and diversity within a microbial community in individual samples [48], whereas beta diversity is a comparative analysis of microbial community composition in different samples. Although no significant difference was recorded for alpha diversity, significant differences in beta diversity were observed between the cecal samples of the two groups, which agreed with previous results showing that *Salmonella* infection can lead to changes in cecal microbiota [21].

The cecal microbial composition of the two groups at both the phylum and genus levels was analyzed using the Wilcoxon test. The results showed that at the phylum level, the relative abundance of *Acidobacteria*, *Campilobacterota*, and *Fusobacteriota* were enriched in group S. The same results were obtained using LEfSe. At the genus level, 18 genera were identified as significantly differential microorganisms by the Wilcoxon test. Among them, *Fusobacterium*, *Helicobacter*, and *Butycicoccus* were identified as marker microorganisms in group S using LEfSe. As we know, *Fusobacterium* has been associated with gastric ulcers in pigs [49] and colon carcinoma in humans [50,51], and may represent a kind of new opportunistic pathogens of chickens worthy of further investigation [52]. In addition, in the species level, *Helicobacter_pullorum* has also been identified as a marker microorganism of group S, which is member of *Campilobacterota* and a well-known zoonotic pathogen [53]. These results revealed that chicks showing higher *S. enteritidis* resistance has lower abundance of pathogenic bacteria in their cecal.

Furthermore, we identified a marker microorganism, *Desulfovibrio_piger*, which was enriched in chicks of group R. *Desulfovibrio_piger*, belonging to *Desulfovibrio spp.*, is a kind of sulfate reducing bacteria, which can functional reducing sulfate to hydrogen sulfide (H_2_S) and plays an important role in intestinal hydrogen and sulfur metabolism. Although H_2_S has been found to have dichotomous effects (stimulatory and inhibitory) on several gastrointestinal processes, it seems to be hazardous at high concentrations but favorable at low concentrations, and the overarching effect of H_2_S appears to be beneficial. For example, H_2_S can attenuate DSS-induced colitis, lessen the shortening of the colon lengths and colonic pathological damages, showing an overall protective effect in colitis via its anti-inflammatory properties [54]. In addition, ATB-429, an H_2_S releasing derivative of mesalamine, exhibits a marked increase in anti-inflammatory activity and potency in a murine model of colitis, as compared to mesalamine, seems promising in the treatment of inflammatory bowel disease [55]. Our results were consistent with these above reports, as our chicks in group R showed higher abundance of *Desulfovibrio_piger* and lower inflammation response at the same time. However, whether *Desulfovibrio_piger* can really help chicks to resist the infection of *S. enteritidis* by producing H_2_S still need to be verified.

## 5. Conclusions

In conclusion, our results confirmed that chicks showed heterogeneous responses to *S. enteritidis* infection, including different degrees of *Salmonella* loads in tissues, different tissue lesion severity, and distinct inflammatory responses. Our findings suggested that enhanced intestinal barrier function and cecal microbiota structure, especially a higher abundance of *Desulfovibrio_piger*, may help chicks resist *S. enteritidis* invasion.

## Figures and Tables

**Figure 1 microorganisms-09-01705-f001:**
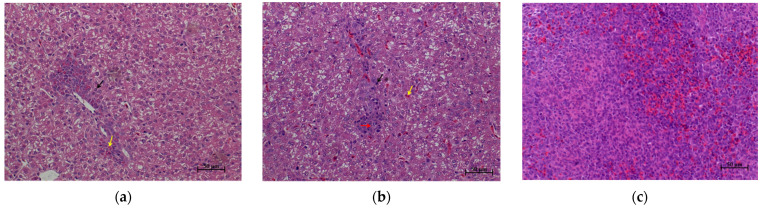
Histopathology of *S. enteritidis*-resistant and *S. enteritidis*-susceptible chicks. (**a**) Representative liver histopathology of *S. enteritidis*-resistant chicks (HE staining); (**b**) Representative liver histopathology of *S. enteritidis*-susceptible chicks (HE staining); (**c**) Representative spleen histopathology of *S. enteritidis*-resistant chicks (HE staining); (**d**) Representative spleen histopathology of *S. enteritidis*-susceptible chicks (HE staining). Original magnification, ×200. Black arrows indicate the lymphocytes, yellow arrows indicate the heterophilic cells, and red arrows indicate the lymphocyte nodules in liver tissue. Scale bar = 50 μm. (**e**) Liver histopathology score of *S. enteritidis*-resistant and -susceptible chicks, *n* = 8 per group, Result of significance test was *p* < 0.05 when marked *.

**Figure 2 microorganisms-09-01705-f002:**
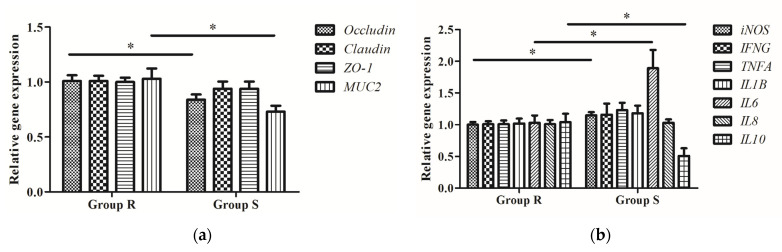
Relative expression of genes coding for tight junction proteins in the jejunum and inflammatory cytokines in cecal tonsil. (**a**) Gene expression levels of *occludin*, *claudin*, *ZO-1*, and *MUC2* in the jejunum of chicks in group R and group S; (**b**) Gene expression levels of *iNOS*, *IFNG*, *TNFA*, *IL1B*, *IL6*, *IL8*, and *IL10* in the cecal tonsil of chicks in group R and group S. Data are presented as means ± SEM (*n* = 8). The asterisk (*) indicates a significant difference between two groups (*p* < 0.05).

**Figure 3 microorganisms-09-01705-f003:**
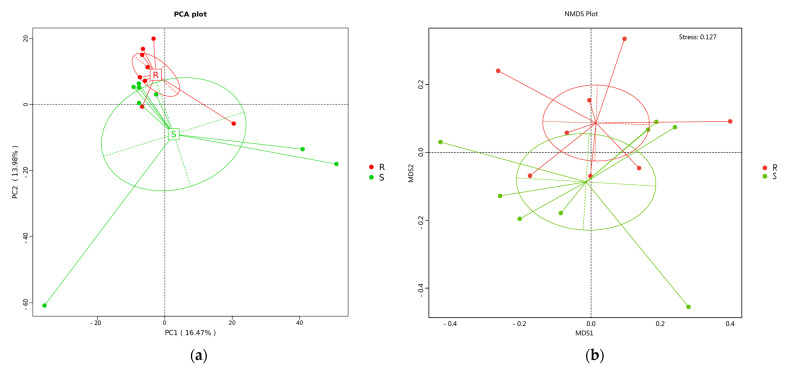
Beta diversity analysis of cecal microbiota (*n* = 8). (**a**) Principal component analysis (PCA). (**b**) Non-metric multidimensional scaling (NMDS). (**c**) Unweighted pair-group method with arithmetic means (UPGMA) clustering tree structure. R = *S. enteritidis*-resistant chicks; S = *S. enteritidis*-susceptible chicks.

**Figure 4 microorganisms-09-01705-f004:**
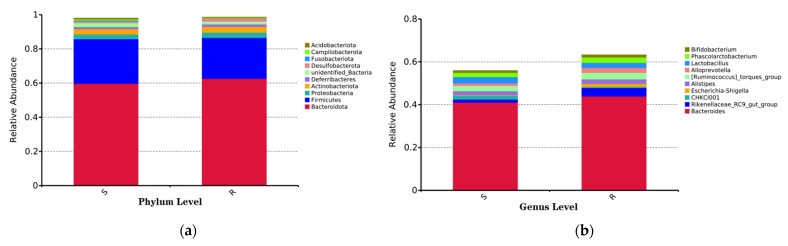
Changes in microbial composition and structure at the phylum and genus levels (*n* = 8). (**a**) Top 10 microbial populations at the phylum level. (**b**) Top 10 microbial populations at the genus level. S = selected *S. enteritidis*-susceptible chicks; R = selected *S. enteritidis*-resistant chicks. (**c**) Complex heatmap of the top 10 microbial populations at the phylum level.

**Figure 5 microorganisms-09-01705-f005:**
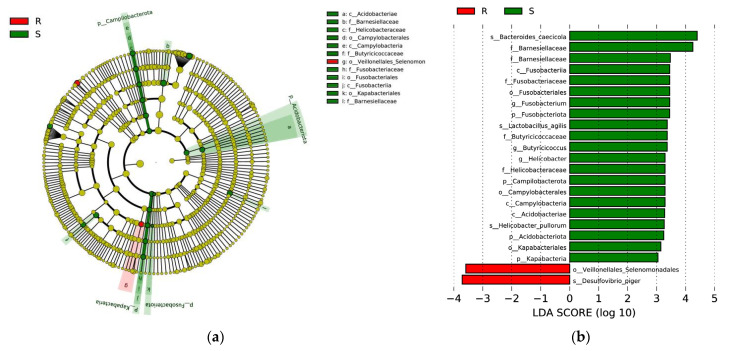
Linear discriminant analysis effect size (LEfSe) identified the most differentially abundant taxa between group S (green bars) and group R (red bars). (**a**) Taxonomic cladogram obtained from LEfSe analysis of 16S rRNA gene sequences. Small circles and shading with different colors in the diagram represent the abundance of those taxa in the respective group. Green circles represent taxa enriched in group S (green legend); Red circles represent taxa enriched in group R (red legend); Yellow circles represent non-significant differences in abundance between two groups. The brightness of each circle is proportional to its effect size. (**b**) Taxa enriched in group S are indicated with a positive LDA score (green), and taxa enriched in group R have a negative LDA score (red). Only those taxa with an LDA value greater than 3 are shown.

**Table 1 microorganisms-09-01705-t001:** Diet composition and nutrient levels (as-fed basis) from 1 to 9 d of age.

Ingredient	%
Corn	55.24
Soybean meal (46%)	36.92
Soybean oil	3.50
Limestone	1.12
Calcium hydrogen phosphate	2.10
Methionine	0.28
Lysine (98%)	0.22
NaCl	0.30
Vitamin premix ^1^	0.03
Mineral premix ^2^	0.20
Choline chloride (70%)	0.09
Total	100.00
Nutrient levels (%) ^3^	
ME (kcal/kg)	2950
CP	21.00
Ca	1.00
Total phosphorus	0.67
Nonphytate phosphorous	0.45
Digestible Lys	1.20
Digestible sulfur-containing amino acid	0.85
Digestible Thr	0.66
Digestible Trp	0.22

^1^ Premix vitamin provided per kilogram of diet: Vitamin A (retinyl palmitate), 8000 IU; vitamin D3 (cholecalciferol), 1000 IU; vitamin E (D, L-α-tocopheryl acetate), 20 IU; vitamin K3 (menadione sodium bisulfate complex), 0.50 mg; vitamin B1, 2.00 mg; vitamin B2, 8.00 mg; vitamin B6, 3.50 mg; vitamin B12 (cobalamin), 10.00 μg; niacin, 35.00 mg; calcium pantothenic, 10.00 mg; folic acid, 0.55 mg; biotin, 0.18 mg. ^2^ Premix mineral provided per kilogram of diet: Fe, 80.00 mg; Mn, 100.00 mg; Zn, 80.00 mg; I, 0.70 mg; Se, 0.30 mg; Cu, 8.00 mg. ^3^ ME was a calculated value, whereas the other nutrient levels were measured values.

**Table 2 microorganisms-09-01705-t002:** Gene-specific primers for related genes.

Gene	GenBank Accession No.	Primer Orientation	Primer Sequence (5′→3′)	Product Size (bp)
*GAPDH*	NM_204305.1	Forward	GCCCAGAACATCATCCCA	137
		Reverse	CGGCAGGTCAGGTCAACA	
*iNOS*	NM_204961.1	Forward	CCTGGAGGTCCTGGAAGAGT	82
		Reverse	CCTGGGTTTCAGAAGTGGC	
*IFNG*	NM_205149.1	Forward	CAAGCTCCCGATGAACGACTT	162
		Reverse	AGTTGAGCACAGGAGGTCAT	
*TNF* *A*	NM_204267.1	Forward	CAGGACAGCCTATGCCAACAAG	114
		Reverse	GGTTACAGGAAGGGCAACTCATC	
*IL1B*	NM_204524.1	Forward	CCGAGGAGCAGGGACTTT	133
		Reverse	AGGACTGTGAGCGGGTGT	
*IL6*	NM_204628.1	Forward	TTTATGGAGAAGACCGTGAGG	106
		Reverse	TGTGGCAGATTGGTAACAGAG	
*IL8*	NM_205498.1	Forward	ATGAACGGCAAGCTTGGAGCTG	233
		Reverse	TCCAAGCACACCTCTCTTCCATCC	
*IL10*	NM_001004414.2	Forward	GCTGAGGGTGAAGTTTGAG	272
		Reverse	CAGGTGAAGAAGCGGTGA	
*occludin*	NM_205128.1	Forward	TCATCGCCTCCATCGTCTAC	141
		Reverse	TCTTACTGCGCGTCTTCTGG	
*claudin*	NM_001013611	Forward	CTGATTGCTTCCAACCAG	140
		Reverse	CAGGTCAAACAGAGGTACAAG	
*ZO-1*	XM_413773	Forward	CTTCAGGTGTTTCTCTTCCTCCTC	131
		Reverse	CTGTGGTTTCATGGCTGGATC	
*MUC2*	NM_001318434.1	Forward	GTGAAGACCCTGATGAAA	219
		Reverse	GTGAACACTGGCGAGAAT	

**Table 3 microorganisms-09-01705-t003:** Body weight, tissue index ^1^, and *Salmonella* loads of *S. enteritidis*-susceptible and -resistant chicks ^2^.

Items	Group S ^3^	Group R ^3^	*p*-Value
BW (g)	229.40 ± 7.47	230.88 ± 5.47	0.875
Liver Index (%)	0.043 ± 0.001 ^a^	0.038 ± 0.001 ^b^	0.006
Spleen Index (%)	0.024 ± 0.002	0.020 ± 0.001	0.158
Liver *Salmonella* loads (log_10_CFU/g)	2.750 ± 0.405 ^a^	1.152 ± 0.435 ^b^	0.018
Spleen *Salmonella* loads (log_10_CFU/g)	4.784 ± 0.100 ^a^	2.491 ± 0.055 ^b^	<0.001

^1^ Tissue index: Percent of tissue weight relative to body weight. ^2^ Results are expressed as means ± SEM, with *n* = 8 per group.^3^ Group S = selected *S. enteritidis*-susceptible chicks; Group R = selected *S. enteritidis*-resistant chicks. ^a,b^ In the same row, values with different letters are significantly different between two groups (*p* < 0.05).

**Table 4 microorganisms-09-01705-t004:** Jejunum morphology of *S. enteritidis*-susceptible and -resistant chicks ^1^.

Items	Group S ^2^	Group R ^2^	*p*-Value
Villus height (μm)	1084.62 ± 35.20	1125.93 ± 90.23	0.683
Crypt depth (μm)	149.56 ± 7.48	131.55 ± 16.28	0.348
Ratio of villus height-to-crypt depth	7.32 ± 0.34	8.94 ± 0.75	0.090
Muscle thickness (μm)	117.86 ± 7.09	118.02 ± 14.37	0.992

^1^ Results are expressed as means ± SEM, with *n* = 8 per group. ^2^ Group S = selected *S. enteritidis*-susceptible chicks; Group R = selected *S. enteritidis*-resistant chicks.

**Table 5 microorganisms-09-01705-t005:** Genera with significant differences between *S. enteritidis*-susceptible and -resistant chicks ^1^.

Taxa	Group S (%) ^2^	Group R (%) ^2^	*p*-Value
*Fusobacterium*	0.5168 ± 0.9028 ^a^	0.1782 ± 0.3837 ^b^	0.043
*Helicobacter*	0.6998 ± 0.6460 ^a^	0.3238 ± 0.5453 ^b^	0.028
*Butyricicoccus*	0.8835 ± 0.5888 ^a^	0.3974 ± 0.4655 ^b^	0.050
*Bryobacter*	0.1440 ± 0.1037 ^a^	0.0426 ± 0.0371 ^b^	0.034
*Acidothermus*	0.1222 ± 0.0886 ^a^	0.0224 ± 0.0251 ^b^	0.026
*unidentified_Chloroplast*	0.0637 ± 0.0814 ^a^	0.0256 ± 0.0469 ^b^	0.039
*NK4A214_group*	0.0560 ± 0.0504 ^a^	0.0214 ± 0.0243 ^b^	0.040
*Marvinbryantia*	0.0685 ± 0.0414 ^a^	0.0278 ± 0.0272 ^b^	0.046
*Burkholderia-Caballeronia-Paraburkholderia*	0.0525 ± 0.0394 ^a^	0.0112 ± 0.0166 ^b^	0.024
*Granulicella*	0.0458 ± 0.0352 ^a^	0.0096 ± 0.0130 ^b^	0.035
*Puia*	0.0218 ± 0.0228 ^a^	0.0035 ± 0.0056 ^b^	0.048
*Oribacterium*	0.0013 ± 0.0036 ^b^	0.0102 ± 0.0162 ^a^	0.028
*unidentified_IMCC26256*	0.0182 ± 0.0134 ^a^	0.0042 ± 0.0067 ^b^	0.037
*Actinospica*	0.0102 ± 0.0092 ^a^	0.0013 ± 0.0024 ^b^	0.021
*Herbinix*	0 ^b^	0.0064 ± 0.0079 ^a^	0.013
*Dyella*	0.0074 ± 0.0079 ^a^	0.0010 ± 0.0019 ^b^	0.037
*Nocardia*	0.0061 ± 0.0056 ^a^	0.0006 ± 0.0018 ^b^	0.017
*Papillibacter*	0.0003 ± 0.0009 ^b^	0.0041 ± 0.0034 ^a^	0.034

^1^ Results are expressed as means ± SEM, with *n* = 8 per group. ^2^ Group S = selected *S. enteritidis*-susceptible chicks; Group R = selected *S. enteritidis*-resistant chicks. ^a,b^ In the same row, values with different letters are significantly different between 2 groups (*p* < 0.05).

## Data Availability

These sequence data have been submitted to the Biotechnology Information (NCBI) Sequence Read Archive databases under accession number PRJNA742372.

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
