# Peer review of "Infection Heterogeneity and Microbiota Differences in Chicks Infected by Salmonella enteritidis"

_microorganisms, 2021, doi:10.3390/microorganisms9081705_

Round 1

Reviewer 1 Report

The manuscript authored by Wu et al, reports the heterogeneity in the susceptibility of genetically homogeneous chickens to Salmonella enteritidis infection. The authors classified chickens as susceptible or resistant based on the level of bacterial load at 24 hr post Salmonella infection and investigated microscopic lesions in the intestine and internal organs, analyzed expression of jejunum tight junction genes and inflammatory genes in the cecal tonsils by qRT-PCR. They also compared the cecal microbiome diversity of susceptible and resistant birds. Based on their findings the authors concluded that Desulfovibrio piger abundance in the intestine of chickens might be a contributing factor to Salmonella resistance between genetically homogeneous population of chickens. The manuscript is very well written and informative. However, before publication the authors should address the following comments

  1. The authors should indicate the cut-off of value of Salmonella load for classifying birds as Salmonella resistant or susceptible?
  2. The authors should provide data on the tight junction gene expression levels as well as intestinal microbial structure in Salmonella non-infected birds. What if the levels of Desulfovibrio piger before Salmonella infection was similar in all the birds, but its level suppressed in the birds classified as susceptible after Salmonella infection for some other reason?
  3. Why did the authors examine body weight, tissue index and Salmonella loads only at 24 hrs post infection? I assume they would have gotten more information if they examined for extended time points!
  4. The authors indicated that the birds were examined for their Salmonella free status before including them in the experimental groups. Have they also been examined for other immunosuppressive pathogens and intestinal pathogens that would facilitate Salmonella colonization? Because heterogeneous response might be observed if some of the birds were sub-clinically infected with any immunosuppressive pathogen.
  5. Intestinal morphology and barrier function was examined in the jejunum while microbiome study was conducted only in the ceca. Why didn’t the authors look at the differences in jejunal microbiota between susceptible and resistant chickens?
  6. Fig 1a & b. Please indicate the lesions with arrows

Author Response

Q1. The authors should indicate the cut-off of value of Salmonella load for classifying birds as Salmonella resistant or susceptible?

A1: Thanks very much for your comments. We supplemented the Salmonella load for all 48 chicks in the revised ‘manuscript-supplementary.doc’ file at ‘Table S1. Tissue Salmonella loads of all slaughtered chicks (n = 48). In our experiment conditions here, the cut-off values for classifying birds as Salmonella resistant or susceptible were selected according to ‘Spleen Salmonella loads (log10CFU/g)’, namely, the cut-off value for Group R was >4.400, and the cut-off value for Group S was < 2.700. According to this two cut-off values, eight chicks with the greatest Salmonella tissue loads were assigned to group S, and eight chicks with the lowest Salmonella tissue loads were assigned to group R. And related descriptions were revised at Line 122-126.

Line 122-126:

According to the result of spleen Salmonella loads (log10CFU/g), we select log10CFU/g > 4.400 as the cut-off value for S.Enteritidis resistant chicks, and log10CFU/g < 2.700 as the cut-off value for S.Enteritidis susceptible chicks. According to the cut-off values, eight chicks were assigned to group S, and eight chicks were assigned to group R (Table S1).

Q2: The authors should provide data on the tight junction gene expression levels as well as intestinal microbial structure in Salmonella non-infected birds. What if the levels of Desulfovibrio piger before Salmonella infection was similar in all the birds, but its level suppressed in the birds classified as susceptible after Salmonella infection for some other reason?

A2: Thanks very much for your comments. However, the aim of this study was trying to compare the infection heterogeneity and cecal microbiota in genetically homogeneous chicks infected by S.Enteritidis, and based on this purpose, we consulted many previous studies and designed the experiment without negative control group. Some references for specific experimental design are as follows: Jacobson, et al., 2018; Wang, et al., 2014; Li, et al., 2010. In addition, we agree that it is of far-reaching significance to study the intestinal microbial structure of different chicks before Salmonella attack, and we are considering further in-depth study.

Jacobson, et al., 2018; Wang, et al., 2014; Li, et al., 2010:

Jacobson, A., Lam, L., Rajendram, M., Tamburini, F., Honeycutt, J., Pham, T., Van Treuren, W., Pruss, K., Stabler, S. R., Lugo, K., et al. A gut commensal-produced metabolite mediates colonization resistance to Salmonella infection. Cell Host Microbe. 2018;24(2):296-307.e7. doi: 10.1016/j.chom.2018.07.002.

Li, X., Swaggerty, C. L., Kogut, M. H., Chiang, H. I., Wang, Y., Genovese, K. J., He, H., Zhou, H. Gene expression profiling of the local cecal response of genetic chicken lines that differ in their susceptibility to Campylobacter jejuni colonization. PLoS One. 2010;5(7):e11827. doi: 10.1371/journal.pone.0011827.

Wang, Y., Lupiani, B., Reddy, S. M., Lamont, S. J., Zhou, H. RNA-seq analysis revealed novel genes and signaling pathway associated with disease resistance to avian influenza virus infection in chickens. Poult Sci. 2014;93(2):485-93. doi: 10.3382/ps.2013-03557.

Q3: Why did the authors examine body weight, tissue index and Salmonella loads only at 24 hrs post infection? I assume they would have gotten more information if they examined for extended time points!

A3: Thanks very much for your comments. In fact, under our experimental conditions, the clinical characteristics of Salmonella Enteritidis infected chicks within 24 h post-inoculation were the most obvious (which can also be found in our previous studies, such as Shi, et al., 2018), and showed very significantly susceptibility differences. In addition, although the susceptibility of Salmonella Enteritidis among chicks was also quite different at 3 days after infection, the intra-group differences of different treatment groups was significantly less than the inter-group differences (these part of data came from another experimental study in the same period of us, but the related article is still in writing, and data are not shown here). Therefore, in this manuscript, we chose the data of time point 24 hours after infection for analysis and elaboration.

One of our previous study:

Shi, S., Wu, S., Shen, Y., Zhang, S., Xiao, Y., He, X., Gong, J., Farnell, Y., Tang, Y., Huang, Y., et al. Iron oxide nanozyme suppresses intracellular Salmonella Enteritidis growth and alleviates infection in vivo. Theranostics. 2018;8(22):6149-6162. doi: 10.7150/thno.29303.

Q4: The authors indicated that the birds were examined for their Salmonella free status before including them in the experimental groups. Have they also been examined for other immunosuppressive pathogens and intestinal pathogens that would facilitate Salmonella colonization? Because heterogeneous response might be observed if some of the birds were sub-clinically infected with any immunosuppressive pathogen.

A4: Thanks very much for your useful advice. However, the interaction among so many different pathogens, or the interaction between gut commensal flora and exogenous pathogens, is a very huge and complex subject. Therefore, researcher always do not take these too complex and unexpected factors into their experiment research, but only excluded their target pathogen for all tested animals. Our study here also excluded only our target pathogen, Salmonella, in all tested chicks, without excluding other pathogens. Some similar previous studies can be seen below. But we can't deny that your suggestion is very meaningful and worthy of further study. 

Some similar previous studies:

Kappala, D., Sarkhel, R., Dixit, S. K., Lalsangpuii, Mahawar, M., Singh, M., Ramakrishnan, S., Goswami, T. K. Role of different receptors and actin filaments on Salmonella Typhimurium invasion in chicken macrophages. Immunobiology. 2018;223(6-7):501-507. doi: 10.1016/j.imbio.2018.01.003.

Yan, G. L., Guo, Y. M., Yuan, J. M., Liu, D., Zhang, B. K. Sodium alginate oligosaccharides from brown algae inhibit Salmonella Enteritidis colonization in broiler chickens. Poult Sci. 2011;90(7):1441-8. doi: 10.3382/ps.2011-01364.

Agunos, A., Ibuki, M., Yokomizo, F., Mine, Y. Effect of dietary beta1-4 mannobiose in the prevention of Salmonella enteritidis infection in broilers. Br Poult Sci. 2007;48(3):331-41. doi: 10.1080/00071660701370442.

Q5: Intestinal morphology and barrier function was examined in the jejunum while microbiome study was conducted only in the ceca. Why didn’t the authors look at the differences in jejunal microbiota between susceptible and resistant chickens?

A5: Thanks very much for your comments. In fact, intestinal morphology and barrier function, can also be regarded as the infection heterogeneity phenotypes of chicks infected by Salmonella Enteritidis. Researches on avian intestinal health is generally carried out in the small intestine (Kwak et al., 2021; Zhang et al., 2021; Kim et al., 2021). Therefore, we chose the jejunum as a representative intestinal segment to compare the intestinal morphology and barrier function of susceptible and resistant birds. In addition, we want to explore the differences of S.Enteritidis susceptibility among different chicks from the perspective of intestinal microbiota, and as the most densely colonized microbial habitat in chicken, the cecum is widely used to study the intestinal microbiota of chicken (Kwak et al., 2021; Zhang et al., 2021; Kim et al., 2021). Therefore, we chose the cecum for 16S analysis to compare the differences of intestinal flora composition between susceptible and non susceptible chickens.

Kwak et al., 2021; Zhang et al., 2021; Kim et al., 2021:

Kwak, M. J., Park, M. Y., Choi, Y. S., Cho, J., Pathiraja, D., Kim, J., Lee, H., Choi, I. G., Whang, K. Y. Dietary sophorolipid accelerates growth by modulation of gut microbiota population and intestinal environments in broiler chickens. J Anim Sci Biotechnol. 2021;12(1):81. doi: 10.1186/s40104-021-00606-x.

Zhang, X., Zhao, Q., Wen, L., Wu, C., Yao, Z., Yan, Z., Li, R., Chen, L., Chen, F., Xie, Z., et al. The effect of the antimicrobial peptide Plectasin on the growth performance, intestinal health, and immune function of yellow-feathered chickens. Front Vet Sci. 2021;8:688611. doi: 10.3389/fvets.2021.688611.

Kim, M. J., Hosseindoust, A., Lee, J. H., Kim, K. Y., Kim, T. G., Chae, B. J. Hot-melt extruded copper sulfate affects the growth performance, meat quality, and copper bioavailability of broiler chickens. Anim Biosci. 2021. doi: 10.5713/ab.21.0030. Epub ahead of print.

Q6: Fig 1a & b. Pleaseindicated the lesions with arrows.

A6: Thanks very much for your comments. In the revised manuscript, we indicated the lesions with arrows in Fig 1a & b.

In addition, you can also see the attachment to check my point-by-point response. 

Thank you again for your patience and kind advise.We have revised according to your suggestion. Sincerely, we very appreciate for your advice to help us for improving our manuscript.

Reviewer 2 Report

General Comments: The manuscript is well written and provides useful information. There is some question as to whether 24 h post-inoculation is adequate for assessing a response to Salmonella invasion and colonization. Addition of a later timepoint allowing for the passage of transient Salmonella may be preferable. However, the manuscript still provides relevant information.

Specific Comments:

L32 The authors should reference the source of the information, not reference an article that referenced another article for that information.

Table 3 Indicate that the P value for spleen Salmonella was <0.0001 or provide the actual p value (not just 0.000)

Figure 1 Provide quantitative data. Simply stating that there were obvious lesions with numerous lymphocyte nodules is not adequate.

L305 Change layers to hens

L305 Replace newborn with day of hatch.

Author Response

Q1: The manuscript is well written and provides useful information. There is some question as to whether 24 h post-inoculation is adequate for assessing a response to Salmonella invasion and colonization. Addition of a later timepoint allowing for the passage of transient Salmonella may be preferable. However, the manuscript still provides relevant information.

A1: Thanks very much for your comments. In fact, under our experimental conditions, the clinical characteristics of Salmonella Enteritidis infected chicks within 24 h post-inoculation were the most obvious (which can also be found in our previous studies, such as Shi, S., et al., 2018), and showed very significantly susceptibility differences. In addition, although the susceptibility of Salmonella Enteritidis among chicks was also quite different at 3 days after infection, the intra-group differences of different treatment groups was significantly less than the inter-group differences (these part of data came from another experimental study in the same period of us, but the related article is still in writing, and data are not shown here). Therefore, in this manuscript, we chose the data of time point 24 hours after infection for analysis and elaboration.

One of our previous study:

Shi, S., Wu, S., Shen, Y., Zhang, S., Xiao, Y., He, X., Gong, J., Farnell, Y., Tang, Y., Huang, Y., et al. Iron oxide nanozyme suppresses intracellular Salmonella Enteritidis growth and alleviates infection in vivo. Theranostics. 2018;8(22):6149-6162. doi: 10.7150/thno.29303.

Q2: L32 The authors should reference the source of the information, not reference an article that referenced another article for that information.

A2: Thanks very much for your comments. Sorry to say that we failed to found the original reference article, so we revised this sentence and its reference article in Line 32-33 and Line 444-446.

Line 32-33:

Salmonella is a major foodborne pathogen of global importance which has led to large numbers of deaths in humans and caused economic losses in animal husbandry [1].

Line 444-446:

[1] Line Shi, S., Wu, S., Shen, Y., Zhang, S., Xiao, Y., He, X., Gong, J., Farnell, Y., Tang, Y., Huang, Y., et al. Iron oxide nanozyme suppresses intracellular Salmonella Enteritidis growth and alleviates infection in vivo. Theranostics. 2018;8(22):6149-6162. doi: 10.7150/thno.29303.

Q3: Table 3 Indicate that the P value for spleen Salmonella was <0.0001 or provide the actual p value (not just 0.000)

A3: Thanks very much for your comments. The P-value for spleen Salmonella in Table 3 has been changed to <0.001.

Q4: Figure 1 Provide quantitative data. Simply stating that there were obvious lesions with numerous lymphocyte nodules is not adequate.

A4: Thanks very much for your comments. In the revised manuscript, we added a liver histopathology score chart in Fig 1e, and supplemented the corresponding descriptions at Line 137-140, Line 210, Line 213, and Line 215-221.

Line 137-140:

Histopathological images of liver or spleen were scored by a pathology professional who did not know the experimental group according to the number of inflammatory cell nodules and the degree of cell degeneration and necrosis. The score from normal to severe lesions was 0 ~ 4.

Line 210:

(Figure 1a, e)

Line 213:

(Figure 1b, e)

Line 215-221:

Figure 1. Histopathology of S.Enteritidis-resistant and S.Enteritidis-susceptible chicks. (a) Representative liver histopathology of S.Enteritidis-resistant chicks (HE staining); (b) Representative liver histopathology of S.Enteritidis-susceptible chicks (HE staining); (c) Representative spleen histopathology of S.Enteritidis-resistant chicks (HE staining); (d) Representative spleen histopathology of S.Enteritidis-susceptible chicks (HE staining). Original magnification, ×200. Black arrows indicate the lymphocytes, yellow arrows indicate the heterophilic cells, and red arrows indicate the lymphocyte nodules in liver tissue. Scale bar = 50 μm. (e) Liver histopathology score of S.Enteritidis-resistant and -susceptible chicks, n = 8 per group.

Q5: L305 Change layers to hens

A5: Thanks very much for your comments. The ‘layers’ in Line 305 of the original manuscript has been modified to the ‘hens’ in Line 320 of the revised manuscript.

Line 320: hens

Q6: L305 Replace newborn with day of hatch.

A6: Thanks very much for your comments. The ‘newborn’ in Line 305 of the original manuscript has been modified to the ‘1-d-old’ in Line 321 of the revised manuscript.

Line 321: 1-d-old

In addition, you can also see my point-by-point response in the attachment.

Thanks again for your patience and kind advise. We hope that our responses and revisions are acceptable. Please do not hesitate to contact me if more revisions are required.

Round 2

Reviewer 1 Report

The authors addressed some of the concerns I have had!